# Pycnogenol-Assisted Alleviation of Titanium Dioxide Nanoparticle-Induced Lung Inflammation via Thioredoxin-Interacting Protein Downregulation

**DOI:** 10.3390/antiox13080972

**Published:** 2024-08-09

**Authors:** Je-Oh Lim, Woong-Il Kim, So-Won Pak, Se-Jin Lee, Changjong Moon, In-Sik Shin, Sung-Hwan Kim, Jong-Choon Kim

**Affiliations:** 1College of Veterinary Medicine and BK21 FOUR Program, Chonnam National University, Gwangju 61186, Republic of Korea; dvmljo@kiom.re.kr (J.-O.L.); dvmwoong@gmail.com (W.-I.K.); sowonp0112@gmail.com (S.-W.P.); xhdhksdl123@naver.com (S.-J.L.); moonc@jnu.ac.kr (C.M.); dvmmk79@gmail.com (I.-S.S.); 2Herbal Medicine Resources Research Center, Korea Institute of Oriental Medicine, Naju 58245, Republic of Korea; 3Jeonbuk Department of Inhalation Research, Korea Institute of Toxicology, Jeongup 56212, Republic of Korea

**Keywords:** titanium dioxide nanoparticle, lung inflammation, pycnogenol, antioxidative effect, thioredoxin-interacting protein

## Abstract

Titanium dioxide nanoparticles (TiO_2_NPs) are used in products that are applied to the human body, such as cosmetics and food, but their biocompatibility remains controversial. Pycnogenol (PYC), a natural extract of pine bark, exerts anti-inflammatory and antioxidant effects. In this study, we investigated whether PYC effectively alleviates pulmonary toxicity induced by airway exposure to TiO_2_NPs, and the beneficial effects of PYC were explained through the analysis of changes to the mechanism of cytotoxicity. TiO_2_NPs induced pulmonary inflammation and mucus production, increased the levels of malondialdehyde, and upregulated thioredoxin-interacting protein (TXNIP) and cleaved-caspase 3 (Cas3) in the lungs of mice. However, PYC treatment reduced the levels of all toxicity markers of TiO_2_NPs and restored glutathione levels. These antioxidant and anti-inflammatory effects of PYC were also demonstrated in TiO_2_NP-exposed human airway epithelial cells by increasing the mRNA levels of antioxidant enzymes and decreasing the expression of TXNIP, cleaved-Cas3, and inflammatory mediators. Taken together, our results showed that PYC attenuated TiO_2_NP-induced lung injury via TXNIP downregulation. Therefore, our results suggest the potential of PYC as an effective anti-inflammatory and antioxidant agent against TiO_2_NP-induced pulmonary toxicity.

## 1. Introduction

Titanium dioxide nanoparticles (TiO_2_NPs) are among the most manufactured nanomaterials in the world and are used in a wide range of cosmetics and food, as well as in the biomedical field [1,2]. As the use cases of TiO_2_NPs expand, the exposure to the human body increases, which may result in unintended absorption. Nanoparticles with small size could penetrate the systemic circulation through the oral or pulmonary routes and translocate other organs such as the kidneys and liver, and exposure to titanium has been reported to cause carcinogenesis, genotoxicity, and immunological disorders, and could be a potential factor involved in systemic toxicity [3]. In particular, TiO_2_NPs are introduced into the human body by inhalation through the respiratory tract. TiO_2_NP exposure damages the lungs by altering the cell cycle of lung epithelial cells, causing pulmonary inflammation and fibrosis [4,5]. In addition, exposure to TiO_2_NPs is the main culprit for exacerbating respiratory diseases such as asthma and pneumonia [6,7,8]. Yu et al. [9] reported that TiO_2_NPs induced endoplasmic reticulum stress-mediated autophagic cell death in human bronchial epithelial cells. Therefore, reducing the effects of TiO_2_NPs on the lungs and alleviating its side effects should improve the quality of life; however, studies focusing on these issues are lacking.

Excessive immune response and reactive oxygen species (ROS) generation underlie the pulmonary toxicity of TiO_2_NPs [10,11,12]. ROS are cell signaling molecules necessary for normal biological processes, but excessive production of ROS can cause damage to multiple organelles, which ultimately interferes with normal cell physiology [11,13]. Accordingly, cells have an antioxidant system to maintain redox homeostasis. Antioxidants possess the capacity to mitigate cellular damage by eliminating ROS. They can be categorized into two groups: non-enzymatic antioxidants, including glutathione and vitamin C, and enzymatic antioxidants, such as glutathione peroxidase (GPx), glutathione reductase (GR), superoxide dismutase (SOD), and catalase [14,15]. Moreover, thioredoxin (Trx), which controls cytoplasmic and mitochondrial ROS, can be converted into reduced and oxidized forms by exchanging hydrogen ions, thus acting as an antioxidant, and its antioxidant activity is regulated by thioredoxin-interacting protein (TXNIP) [16]. TXNIP functions as an important mediator in various cell signaling pathways such as apoptosis and the inflammatory response [16], and these pathways are also key mechanisms of action of nanomaterials [17,18]. Therefore, TXNIP is a key protein involved in the mechanism of action of TiO_2_NPs. However, the underlying mechanistic basis has not yet been fully elucidated.

Pycnogenol (PYC), a natural extract of pine bark, has beneficial health functions by supporting immunity and reducing the progression of chronic obstructive pulmonary disease (COPD) [19]. In addition, PYC has been reported to have clinical pharmacological effects targeting the lungs, as it not only alleviates asthma but also inhibits pulmonary fibrosis and has anticancer and anti-inflammatory effects [20,21,22]. The main components of PYC are oligomeric procyanidins, flavonoids, and polyphenols, which have strong antioxidant capacity [23,24]. Moreover, PYC modulates apoptosis by inhibiting inflammatory responses and oxidative stress through antioxidant and free radical scavenging properties [25,26,27]. The anti-inflammatory properties of PYC were known to involve nuclear factor kappa B and mitogen-activated protein kinase signaling, and it suppressed asthmatic airway inflammation by modulating inflammatory mediators such as interleukin (IL)-4, -5, and -13, and inhibiting mucin-5AC in goblet cells [19,28,29]. However, studies on the pharmacological efficacy of PYC in pulmonary toxicity induced by TiO_2_NPs exposure are still lacking, and the molecular signaling of the anti-inflammatory efficacy of PYC in association with TXNIP has not yet been elucidated.

In this study, we performed a comprehensive assessment of TiO_2_NP-induced pulmonary toxicity. Moreover, the antioxidant and anti-inflammatory effects of PYC in pulmonary inflammation generated by TiO_2_NPs were examined, and the mechanism of action of PYC was explored by analyzing TXNIP and its downstream signaling.

## 2. Materials and Methods

### 2.1. Nanoparticles

TiO_2_NPs were obtained from Sigma-Aldrich (catalog number 637254, St. Louis, MO, USA), with a particle size smaller than 25 nm. The samples were dissolved in phosphate-buffered saline (PBS) and subjected to sonication using a VCX-130 instrument (Sonics and Materials, Newtown, CT, USA) for 3 min. The sonication parameters were set to 130 W power, 20 kHz frequency, and a pulse ratio of 59/1. This pre-treatment step was performed before analysis or experimentation.

### 2.2. Test Article and Experimental Procedure for In Vivo Experiments

Pycnogenol^®^ (US patent # 4,698,360), extracted exclusively from the bark of French maritime pine, was donated by Horphag Research Ltd. (Route de Belis, France). Female BALB/c mice, aged six weeks, that were free from specific pathogens were procured from Samtako (Osan, Republic of Korea). These mice were then subjected to a quarantine period and allowed to acclimate for 7 days. Animal experiments were conducted in accordance with the National Institute Health Guidelines for the Care and Use of Laboratory Animals. The experimental protocols involving animals (CNU IACUC-YB-2021-92) were approved by the Institutional Animal Care and Use Committee of Chonnam National University.

The mice were allocated into five groups using randomization, with each group consisting of six animals. These groups were identified as vehicle control (VC), TiO_2_NPs (in which only TiO_2_NPs were administered), dexamethasone (DEX) (in which TiO_2_NPs were administered with a 1 mg/kg dose of DEX; positive control), PYC 10 (in which TiO_2_NPs were administered with a 10 mg/kg dose of PYC, sourced from Horphag Research in Le Sen, France), and PYC 20 (in which TiO_2_NPs were administered with a 20 mg/kg dose of PYC). DEX and PYC were solubilized in PBS and afterward fed to the mice via oral route for 2 weeks. Intranasal administration of TiO_2_NPs was performed on mice on days 1, 7, and 13. The dosage used was 20 mg/kg in 50 µL of PBS, and the VC group received an intranasal instillation of 50 μL of PBS. The mice were administered by intranasal drop instillation with a volume of 50 μL using a micropipette under light anesthesia induced with Zoletil 50, and TiO_2_NPs prepared in PBS were continuously mixed using a vortex mixer until the end of administration.

### 2.3. Collection of Bronchoalveolar Lavage Fluid (BALF) and Cell Counting

The mice were euthanized 48 h following the final instillation by an intraperitoneal dose of Zoletil 50. A tracheostomy was also performed: the lungs were subjected to a tracheal cannula through which 0.7 mL of cold PBS was administered and afterward withdrawn to collect BALF. This procedure was performed twice. The inflammatory cells present in BALF were evaluated as described by Lim et al. [18].

### 2.4. Cytokine Assays

The concentrations of cytokines, including tumor necrosis factor-α (TNF-α), IL-1β, and IL-6, in BALF were quantified using ELISA (enzyme-linked immunosorbent assay) kits obtained from BD Biosciences (San Jose, CA, USA). The manufacturer’s instructions were followed during the experimental procedure. Absorbance was measured using an ELISA reader (Bio-Rad Laboratories, Hercules, CA, USA).

### 2.5. Histopathology and Immunohistochemistry (IHC)

Lung tissue was treated with 4% (*v*/*v*) paraformaldehyde for 48 h. The tissues underwent paraffin embedding, followed by sectioning at a thickness of 4 μm. Airway inflammation and mucus production were assessed using H&E staining (Sigma-Aldrich, St. Louis, MO, USA) and PAS (periodic acid-Schiff) solution (IMEB, San Marcos, CA, USA), respectively. Using a light microscope, each slide was manually assessed by histopathologists blinded to the treatment groups as previously described [30]. The tissues were also prepared for IHC as described by Kim et al. [31]. Protein expression was determined using anti-TXNIP (diluted at 1:200; Novus Biologicals, Littleton, CO, USA) and anti-cleaved-caspase 3 (Cas3; diluted at 1:200; Cell signaling, Danvers, MA, USA) antibodies. An image analyzer (IMT i-Solution Software version 21.1, Vancouver, BC, Canada) was used to quantitatively assess airway inflammation, mucus production, and protein expression.

### 2.6. Western Blot Analysis

To measure the level of protein expression, we conducted immunoblotting as described by Jeong et al. [32]. The primary antibodies include anti-TXNIP (Novus Biologicals), anti-β-actin (β-act; Cell Signaling), and anti-cleaved-caspase 3 (Cas3; Cell Signaling). The densitometric analysis of expression was conducted using the Chemi-Doc system (Bio-Rad Laboratories, CA, USA).

### 2.7. Malondialdehyde (MDA) and Glutathione Assays

MDA levels, which serve as an indicator of oxidative stress in cellular and tissue environments, were determined in lung tissue samples using a thiobarbituric acid reactive substances assay kit (Cayman Chemical, Ann Arbor, MI, USA) following the manufacturer’s instructions. The levels of reduced and oxidized glutathione in lung tissue were quantified using a glutathione assay kit (Cayman Chemical, Ann Arbor, MI, USA) by an enzymatic recycling method. Absorbance was determined using an ELISA reader (Bio-Rad Laboratories).

### 2.8. Cell Culture

The NCI-H292 human airway epithelial cell line was acquired from the ATCC (Manassas, VA, USA). The cells were grown in RPMI 1640 medium (WELGENE, Gyeongsan, Republic of Korea) supplemented with 10% FBS and antibiotics. The incubation was carried out in a humidified space at 37 °C with 5% CO_2_. The cells underwent a period of serum starvation for 1 h before use.

### 2.9. Cell Viability Assay

Cell viability was assessed using the EZ-Cytox cell viability assay kit (DAELIL Lab, Seoul, Republic of Korea). NCI-H292 cells were distributed into 96-well plates at a density of 4 × 10^4^ cells per well. Following a 24 h period, fresh medium was introduced along with different concentrations of PYC (5, 10, 20, 40, and 80 µg/mL), and the culture plates were incubated for a further 24 h. Subsequently, live cells were identified by adding 10 µL of the kit solution to each well and incubating for 4 h. The absorbance at 450 nm was determined using an ELISA reader (Bio-Rad Laboratories).

### 2.10. RNA Extraction and qRT-PCR

RNA extraction was performed using the HiGene Total RNA Prep Kit (Biofact, Daejeon, Republic of Korea) and reverse transcription of the extracted RNA into cDNA using a cDNA kit (Qiagen, Hilden, Germany). To measure the mRNA expression levels of proinflammatory cytokines, qRT-PCR was performed as described by Lim et al. [18]. Specific primers used in qRT-PCR experiments are listed in Table 1. A Real-Time PCR Detection System (Bio-Rad Laboratories) was used for quantitative analysis.

### 2.11. Double Immunofluorescence and Confocal Microscope

Double immunofluorescence was performed as described by Kim et al. [33]. The primary antibodies used were anti-TXNIP (diluted 1:200; Novus Biologicals) and anti-cleaved-Cas3 (diluted 1:200; Cell Signaling). Fluorescence images were acquired using a confocal microscope (Leica Microsystems, Heidelberg, Germany).

### 2.12. Statistical Analysis

The data are presented as the mean value accompanied by the standard deviation (SD). The statistical significance of the data was assessed by using ANOVA, followed by Dunnett’s test for multiple comparisons. Statistical significance was attributed to *p*-values that were less than 0.05.

## 3. Results

### 3.1. PYC Reduces the Number of Inflammatory Cells in the BALF

The experimental group exposed to TiO_2_NPs exhibited a notably greater count of inflammatory cells in the BALF in comparison to the VC group (Figure 1). This increase in inflammatory cell count was mostly attributed to a substantial rise in the numbers of neutrophils and macrophages. A notable reduction in the numbers of neutrophils and macrophages was observed in mice that were provided with DEX compared to the group treated with TiO_2_NPs. Furthermore, a notable decrease in inflammatory cell counts was observed, specifically neutrophils and macrophages, in mice treated with PYC in comparison to the group exposed to TiO_2_NPs. Additionally, the extent of these reductions in the PYC groups was dependent on the dosage administered (Figure 1).

### 3.2. PYC Decreases Proinflammatory Cytokines in the BALF

Compared to VC mice, TiO_2_NP-exposed animals demonstrated significantly increased levels of TNF-α in their BALF (Figure 2A). Consistent with the observed increase in TNF-α levels, the concentrations of IL-1β and IL-6 in BALF were elevated in the group exposed to TiO_2_NPs (Figure 2B,C). In contrast, the DEX group exhibited notable reductions in the levels of IL-1β, TNF-α, and IL-6 in comparison to the mice treated with TiO_2_NPs. Furthermore, the group exposed to TiO_2_NPs had elevated levels of IL-1β, TNF-α, and IL-6, which were dramatically decreased after treatment with PYC. Notably, the PYC high-dose group showed a more pronounced decrease in cytokine levels (Figure 2A–C).

### 3.3. PYC Ameliorates Airway Inflammation and Mucous Production in TiO_2_NP-Exposed Mice

The groups of mice that were exposed to TiO_2_NPs exhibited a notable aggregation of inflammatory cells in the vicinity of the alveoli and bronchi (Figure 3). The number of inflammatory cells in the group exposed to TiO_2_NPs was significantly higher in comparison to that of the VC group. In contrast, the DEX group exhibited a marked decrease in lung inflammation compared to the TiO_2_NP group. Similarly, the PYC groups displayed a trend comparable to that of the DEX group in terms of reduced inflammation (Figure 3). In mice exposed to TiO_2_NPs, mucus production increased with inflammatory cell infiltration, and these changes were significant compared to the VC group. However, increased mucus production by TiO_2_NP exposure was reduced in the PYC groups, and this tendency was also observed in the DEX group (Figure 4).

### 3.4. PYC Reduces ROS Production in the Lungs of TiO_2_NP-Exposed Mice

To ascertain the potential of PYC in reducing ROS levels in the lungs of mice exposed to TiO_2_NPs, we performed MDA assays. MDA levels exhibited a substantial increase in the group exposed to TiO_2_NPs, measuring approximately twice as high as those of the group treated with PBS only. In contrast, in the DEX group, MDA levels exhibited a significant reduction compared to those of the TiO_2_NPs group, with a tendency to recover to a level comparable to that of the VC group. Furthermore, an MDA level pattern similar to that in the DEX group was observed in the PYC groups. Additionally, a more pronounced decrease was observed in the group given a high dose of PYC (Figure 5A). In contrast, glutathione levels in each group showed the opposite tendency to that of MDA levels. Glutathione levels in the lungs of the TiO_2_NPs group were significantly reduced compared to those in the VC group, and the DEX and PYC groups displayed recovery outcomes that were comparable to those of the VC group (Figure 5B).

### 3.5. PYC Inhibits TXNIP and Apoptotic Protein Expression in the Lungs of TiO_2_NP-Exposed Mice

Using IHC, the expression levels of TXNIP and cleaved-Cas3 in lung tissues were assessed in response to treatments with TiO_2_NP, DEX, and PYC. The TiO_2_NP group exhibited considerably increased expression of TXNIP and cleaved-Cas3 in comparison to the VC group. In contrast to the TiO_2_NP group, DEX-treated mice showed lower levels of cleaved-Cas3 and TXNIP expression. This tendency was also observed in PYC-treated mice, showing that the expression of TXNIP and cleaved-Cas3 increased by TiO_2_NP was effectively reduced in the PYC groups (Figure 6A,B). When the expression of TXNIP and cleaved-Cas3 was measured using Western blotting, the same trend was obtained as in the IHC analysis (Figure 6C,D).

### 3.6. PYC Inhibits mRNA Expression of Proinflammatory Cytokines and Activates mRNA Expression of Antioxidant Enzymes in TiO_2_NP-Induced NCI-H292 Cells

Cell viability assay results showed that the cells treated with 10 µg/mL PYC or less had more than 85% viability; in contrast, cell viability was reduced by nearly half when treated with 40 µg/mL PYC (Figure 7A). The mRNA levels of the proinflammatory cytokines TNF-α, IL-1β, and IL-6 were considerably upregulated in NCI-H292 cells treated with TiO_2_NPs compared to those in untreated NCI-H292 cells. In NCI-H292 cells treated with PYC, the mRNA expression of proinflammatory mediators was markedly decreased compared to those of cells treated with TiO_2_NP only (Figure 7B–D). Furthermore, PYC treatment increased the mRNA expression of antioxidant enzymes in a concentration-dependent manner compared to those of cells treated with TiO_2_NP only (Figure 7E–G).

### 3.7. PYC Inhibits TXNIP and Apoptotic Protein Expression in TiO_2_NP-Induced NCI-H292 Cells

Exposure to TiO_2_NPs markedly increased the expression of TXNIP, and PYC treatment suppressed the elevated expression of TXNIP in a concentration-dependent manner (Figure 8A,B). Moreover, a similar pattern was observed in the expression of cleaved-Cas3, an apoptosis indicator protein. The expression of cleaved-Cas3 was significantly enhanced after exposure to TiO_2_NPs in comparison to non-exposed NCI-H292 cells. Furthermore, it was observed that the administration of PYC effectively reduced the increased expression of cleaved-Cas3 (Figure 8A,B).

## 4. Discussion

TiO_2_NPs are being used more frequently in a diverse range of industrial applications. Recently, exposure to TiO_2_NPs has been shown to induce oxidative stress and an inflammatory response in the respiratory tract [12]. Nevertheless, the toxic effects and underlying mechanisms of TiO_2_NP toxicity have not yet been completely elucidated. This study was performed to characterize the toxic effects of TiO_2_NPs in mouse lungs and human airway epithelial cells. Moreover, the antioxidant and anti-inflammatory effects of PYC in TiO_2_NP-induced pulmonary inflammation were investigated, and the mechanism of action of PYC was explored by analyzing TXNIP expression and its downstream signaling. TiO_2_NP exposure significantly increased inflammatory mediators, ROS, and mucus production in the lungs of mice, accompanied by an activation of TXNIP expression and apoptotic signaling. However, PYC treatment alleviated TiO_2_NP-induced pulmonary toxicity by suppressing the expression of TXNIP, and these protective effects of PYC were confirmed in vitro.

Inflammation is a protective response of the host to harmful substances, and it is an essential program for removing harmful foreign substances, repairing damaged tissues, and maintaining homeostasis. However, when this process is disturbed by an excessive inflammatory response, the tissue is damaged [34]. Moreover, as alveolar macrophages fail to eliminate invading pathogens and foreign substances, cytokines and chemokines are released to attract neutrophils [35]. In this study, proinflammatory cytokine levels increased owing to immune system activation by TiO_2_NPs being recognized as a foreign substance, and the migration and accumulation of inflammatory cells increased, resulting in lung injury. However, these lung injury-related indicators were lowered in the lungs of PYC-treated mice, and PYC reduced the levels of the proinflammatory cytokines IL-1β, IL-6, and TNF-α in TiO_2_NP-exposed human bronchial epithelial cells. According to recent articles [19,36], PYC exerts an anti-inflammatory effect by reducing the release of TNF-α, IL-6, and IL-1β and inhibiting inflammatory cell infiltration, and these effects ameliorate the lung dysfunction caused by COPD. Furthermore, PYC has an inhibitory effect on mucus production by reducing the expression of mucin 5AC mRNA [19]. Taken together, our results suggest that the anti-inflammatory effects of PYC can prevent lung damage from TiO_2_NP exposure.

When a cell is subjected to excessive stress, it generates large amounts of ROS, which have a strong chemical affinity to other substances in the body, attacking the membrane of cells or organs, thereby impairing cellular functions and leading to cell death. In normal conditions, ROS are maintained at low levels, but when they are increased by external stress, oxidative stress and apoptosis occur [37,38]. Inflammation and ROS are importantly intertwined in cellular physiological processes. TNF-α and IL-1β increase ROS generation and cause mitochondrial dysfunction, resulting in redox imbalance [11,39]. In this study, TiO_2_NP exposure increased MDA levels and cleaved-Cas3 expression and decreased glutathione levels in the lungs of mice. Moreover, TXNIP expression increased in mice exposed to TiO_2_NPs. In a recent study, TiO_2_NP exposure was shown to elevate the expression of cleaved-Cas3 and Bax by increasing the expression of TXNIP, which regulates the Trx antioxidant system, and the downregulation of TXNIP reduces pulmonary toxicity and apoptosis induced by TiO_2_NPs [18]. Furthermore, our results showed that PYC treatment suppressed TiO_2_NP adverse effects, such as increased expression of TXNIP, cleaved-Cas3, and MDA, and restored glutathione to normal levels. In TiO_2_NP-exposed cells, PYC treatment increased the expression of antioxidant enzymes and decreased the expression of TXNIP. PYC contains polyphenols and functions as a free radical scavenger, and the free radical scavenger N-acetyl cysteine and the antioxidant polyphenols inhibit TXNIP activity [35,40]. Taken together, these findings suggest that PYC is involved in TXNIP regulation and inhibits TXNIP-ROS-apoptotic signaling activated by TiO_2_NPs.

In this study, the pharmacological efficacy of PYC in TiO_2_NP-induced lung inflammation was evaluated, and PYC effectively alleviated the pulmonary inflammatory response by inhibiting TiO_2_NP-induced TXNIP-ROS-apoptosis. Although potential toxicity of PYC was not evaluated in our data, it has been reported in many literature works that it has protective effects not only in the respiratory tract but also in other organs such as the heart, brain, kidney, and liver [24,41,42]. In addition, there are also literature works that set higher doses of PYC than those used in our study [28,41]; therefore, the adverse effects of PYC itself in this study were expected to be minimal. To complement the limitations of this study, future studies are suggested to evaluate the potential adverse effects of PYC itself as well as the anti-inflammatory effects of PYC in representative respiratory diseases such as asthma and COPD, thereby elucidating new molecular mechanisms of the pharmacological efficacy of PYC.

## 5. Conclusions

TiO_2_NP exposure in mouse lungs and human bronchial epithelial cells increased the inflammatory response, ROS generation, and apoptosis by increasing TXNIP expression, and PYC reduced TXNIP expression, thereby alleviating the pulmonary toxicity of TiO_2_NPs (Figure 9). Therefore, our results suggest the potential of PYC as an effective anti-inflammatory agent and antioxidant in TiO_2_NP-induced pulmonary toxicity.

## Figures and Tables

**Figure 1 antioxidants-13-00972-f001:**
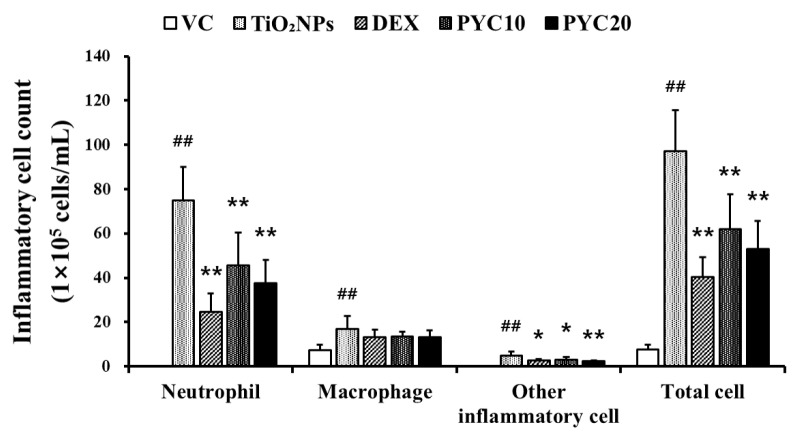
Effects of pycnogenol treatment on inflammatory cell counts in the BALF. VC, PBS intranasal instillation; TiO_2_NP, TiO_2_NP intranasal instillation; DEX, TiO_2_NP intranasal instillation + dexamethasone administration (1 mg/kg); PYC10 and 20, TiO_2_NP intranasal instillation + pycnogenol administration (10 and 20 mg/kg, respectively). Data are represented as means ± SD, *n* = 6. ^##^ Statistical significance from the VC group, *p* < 0.01; *, ** Statistical significance from the TiO_2_NP group, *p* < 0.05 and <0.01, respectively.

**Figure 2 antioxidants-13-00972-f002:**
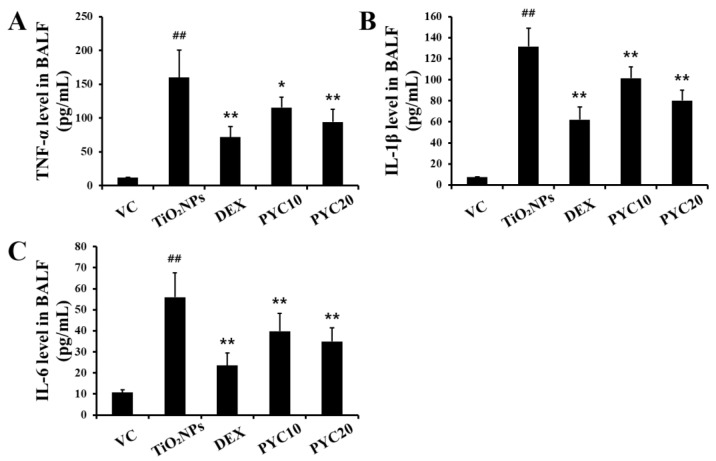
Effects of pycnogenol treatment on proinflammatory cytokines levels in the BLAF. (**A**) TNF-α. (**B**) IL-1β. (**C**) IL-6. VC, PBS intranasal instillation; TiO_2_NP, TiO_2_NP intranasal instillation; DEX, TiO_2_NP intranasal instillation + dexamethasone administration (1 mg/kg); PYC10 and 20, TiO_2_NP intranasal instillation + pycnogenol administration (10 and 20 mg/kg, respectively). Data are represented as means ± SD, *n* = 6. ^##^ Statistical significance from the VC group, *p* < 0.01; *, ** Statistical significance from the TiO_2_NP group, *p* < 0.05 and <0.01, respectively.

**Figure 3 antioxidants-13-00972-f003:**
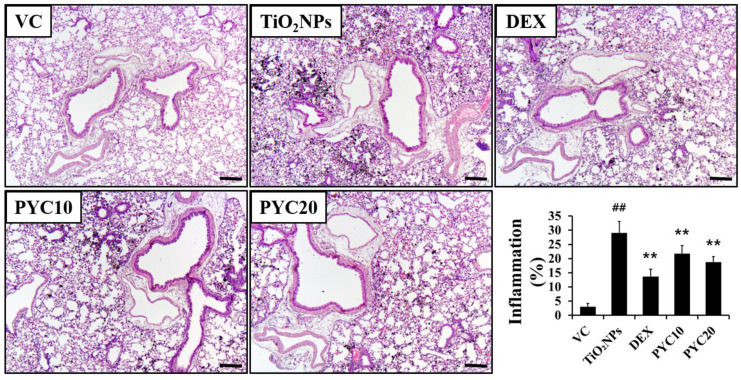
Effects of pycnogenol treatment on inflammatory responses in the lungs. Lung tissue is stained with H&E stain (×100). VC, PBS intranasal instillation; TiO_2_NP, TiO_2_NP intranasal instillation; DEX, TiO_2_NP intranasal instillation + dexamethasone administration (1 mg/kg); PYC10 and 20, TiO_2_NP intranasal instillation + pycnogenol administration (10 and 20 mg/kg, respectively). Data are represented as means ± SD, *n* = 6. ^##^ Statistical significance from the VC group, *p* < 0.01; ** Statistical significance from the TiO_2_NP group, *p* < 0.01. Bar = 100 μm.

**Figure 4 antioxidants-13-00972-f004:**
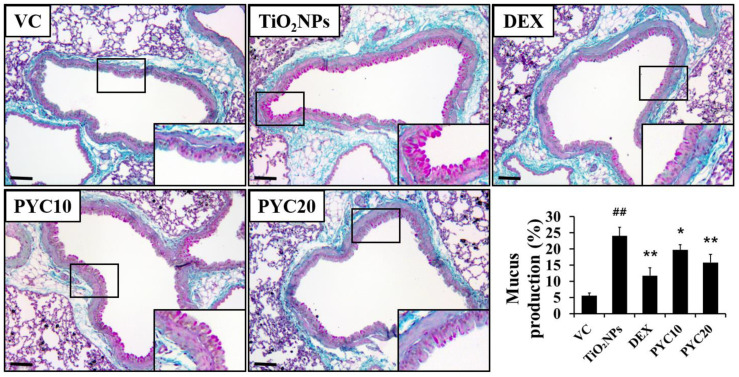
Effects of pycnogenol treatment on mucus production in the lungs. Lung tissue is stained with PAS stain (×200). VC, PBS intranasal instillation; TiO_2_NP, TiO_2_NP intranasal instillation; DEX, TiO_2_NP intranasal instillation + dexamethasone administration (1 mg/kg); PYC10 and 20, TiO_2_NP intranasal instillation + pycnogenol administration (10 and 20 mg/kg, respectively). Data are represented as means ± SD, *n* = 6. ^##^ Statistical significance from the VC group, *p* < 0.01, respectively; *, ** Statistical significance from the TiO_2_NP group, *p* < 0.05 and <0.01, respectively. Bar = 50 μm.

**Figure 5 antioxidants-13-00972-f005:**
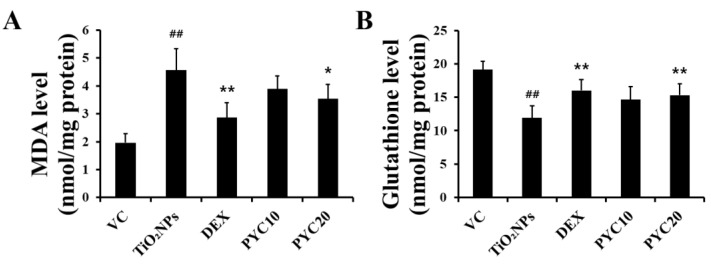
Effects of pycnogenol treatment on levels of MDA and glutathione in TiO_2_NP-exposed mice. (**A**) MDA level in the lungs. (**B**) Glutathione level in the lungs. VC, PBS intranasal instillation; TiO_2_NP, TiO_2_NP intranasal instillation; DEX, TiO_2_NP intranasal instillation + dexamethasone administration (1 mg/kg); PYC10 and 20, TiO_2_NP intranasal instillation + pycnogenol administration (10 and 20 mg/kg, respectively). Data are represented as means ± SD, *n* = 6. ^##^ Statistical significance from the VC group, *p* < 0.01; *, ** Statistical significance from the TiO_2_NP group, *p* < 0.05 and <0.01, respectively.

**Figure 6 antioxidants-13-00972-f006:**
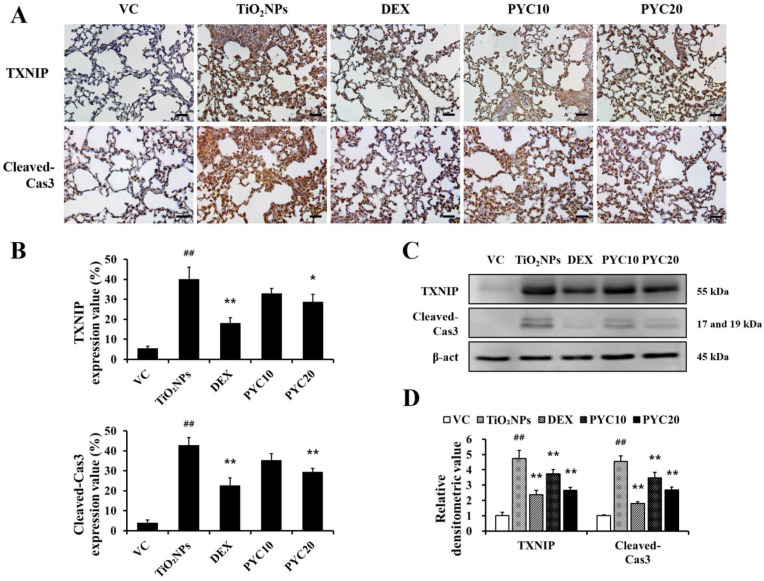
Effects of pycnogenol treatment on the expression of TXNIP and cleaved-Cas3 in TiO_2_NP-exposed mice. (**A**) Expression of TXNIP and cleaved-Cas3 (×400, alveolar). (**B**) Quantitation of TXNIP- and cleaved-Cas3-positive cell. (**C**) Western blotting. (**D**) Relative densitometric values. VC, PBS intranasal instillation; TiO_2_NP, TiO_2_NP intranasal instillation; DEX, TiO_2_NP intranasal instillation + dexamethasone administration (1 mg/kg); PYC10 and 20, TiO_2_NP intranasal instillation + pycnogenol administration (10 and 20 mg/kg, respectively). Data are represented as means ± SD, *n* = 6. ^##^ Statistical significance from the VC group, *p* < 0.01; *, ** Statistical significance from the TiO_2_NP group, *p* < 0.05 and <0.01, respectively. Bar = 25 μm.

**Figure 7 antioxidants-13-00972-f007:**
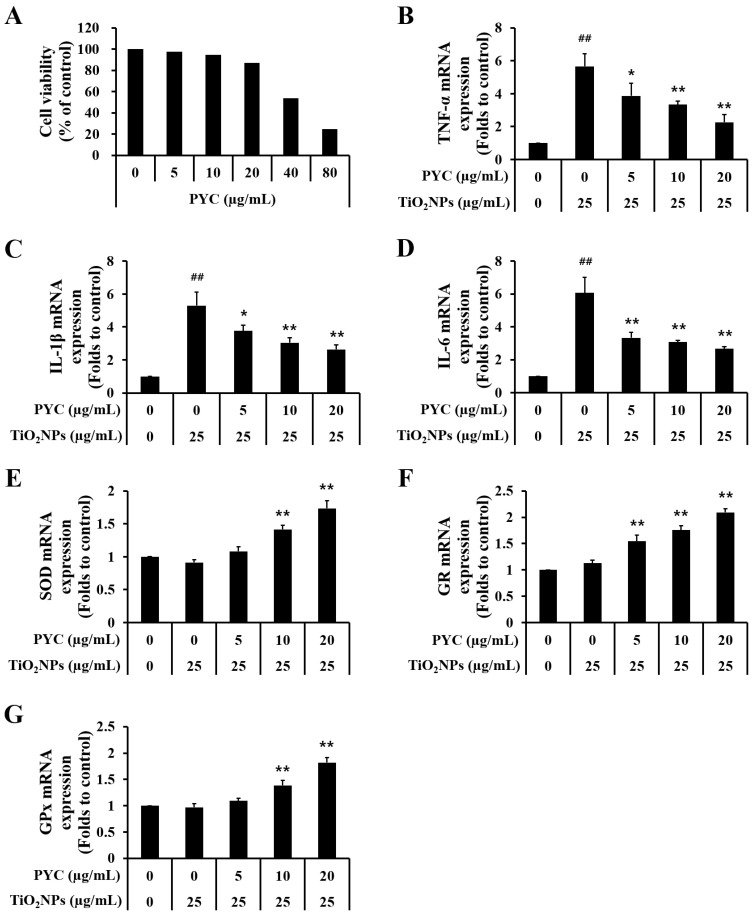
Effects of pycnogenol treatment on mRNA expression of proinflammatory cytokines and antioxidant enzymes in TiO_2_NP-induced NCI-H292 cells. (**A**) Cell viability. (**B**) TNF-α. (**C**) IL-1β. (**D**) IL-6. (**E**) SOD. (**F**) GR. (**G**) GPx. Data are represented as means ± SD, *n* = 3. ^##^ Statistical significance from non-induced NCI-H292 cells, *p* < 0.01; *, ** Statistical significance from TiO_2_NP-induced NCI-H292 cells, *p* < 0.05 and <0.01, respectively.

**Figure 8 antioxidants-13-00972-f008:**
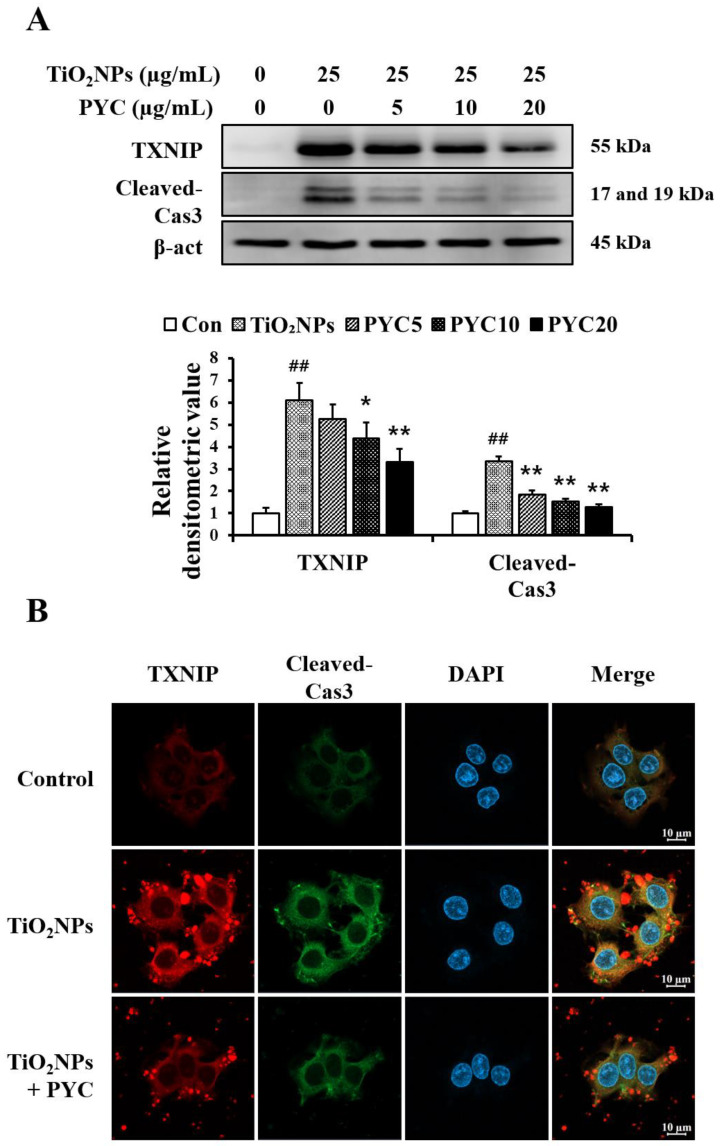
Effects of pycnogenol treatment on the expression of TXNIP and cleaved-Cas3 in TiO_2_NP-induced NCI–H292 cells. (**A**) Western blotting. (**B**) Double-immunofluorescence staining. Bar = 10 μm. Data are represented as means ± SD, *n* = 3. ^##^ Statistical significance from non-induced NCI-H292 cells, *p* < 0.01; *, ** Statistical significance from TiO_2_NP-induced NCI-H292 cells, *p* < 0.05 and <0.01, respectively.

**Figure 9 antioxidants-13-00972-f009:**
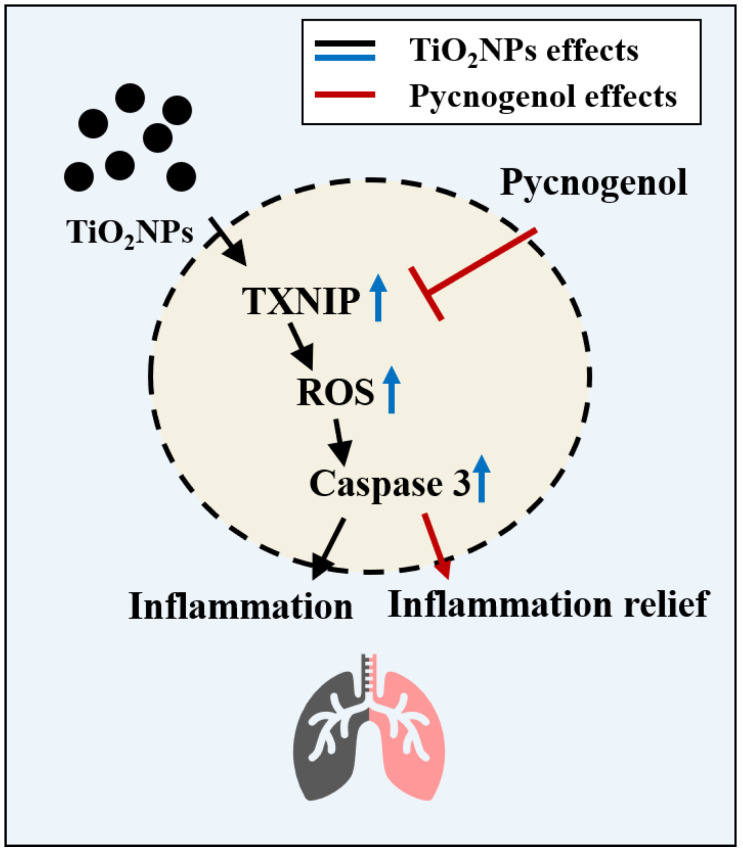
Proposed signaling pathway for the mitigating effects of PYC on TiO_2_NP-induced apoptosis in the lung of asthmatic mice. The blue arrows indicate TiO_2_NP-induced upregulated expression.

**Table 1 antioxidants-13-00972-t001:** Primer sequences for qRT-PCR.

Target Genes		Sequence (5′ → 3′)	T_m_ °C	Accession Number
TNF-α	Forward	CAA AGT AGA CCT GCC CAG AC	59.3	NM_000594
	Reverse	GAC CTC TCT CTA ATC AGC CC	59.3	
IL-6	Forward	ATG CAA TAA CCA CCC CTG AC	57.3	NM_000600
	Reverse	ATC TGA GGT GCC CAT GCT AC	59.3	
IL-1β	Forward	AGC CAG GAC AGT CAG CTC TC	61.4	NM_000576
	Reverse	ACT TCT TGC CCC CTT TGA AT	55.2	
GR	Forward	TTC CAG AAT ACC AAC GTC AAA GG	58.8	NM_000637
	Reverse	GTT TTC GGC CAG CAG CTA TTG	59.8	
SOD	Forward	GGT GGG CCA AAG GAT GAA GAG	61.7	NM_000454
	Reverse	CCA CAA GCC AAA CGA CTT CC	59.3	
GPx	Forward	CAG TCG GTG TAT GCC TTC TCG	61.7	NM_000581
	Reverse	GAG GGA CGC CAC ATT CTC G	60.9	
GAPDH	Forward	CAA AAG GGT CAT CAT CTC TG	55.2	NM_002046
	Reverse	CCT GCT TCA CCA CCT TCT TG	59.3	

TNF-α, tumor necrosis factor-alpha; IL-6, interleukin-6; IL-1β, interleukin-1β; GAPDH, glyceraldehydes-3-phosphate dehydrogenase; GR, glutathione reductase; SOD, superoxide dismutase; GPx, glutathione peroxidase; T_m_, melting temperature of primer; and T_A_, annealing temperature in PCR.

## Data Availability

All the data supporting the results were shown in the paper and can be obtained from the corresponding author.

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
