# Peer review of "Pycnogenol-Assisted Alleviation of Titanium Dioxide Nanoparticle-Induced Lung Inflammation via Thioredoxin-Interacting Protein Downregulation"

_antioxidants, 2024, doi:10.3390/antiox13080972_

Round 1

Reviewer 1 Report

The manuscript describes PYC effect on TiO2NPs induced lung inflammation. It is overall well written and interesting. I have two major comments. It seems that Dex is more effective than PYC. I assume Dex was used as a positive control, however, there is no explanation about Dex in the main text (why Dex is examined). The authors need to explain. The other comment, the authors had better add the figure that shows the relationships of the molecules in the text, The authors examined not only inflammation but also oxidative stress and apoptosis. The results are interesting but complicated. Therefore, the figure that shows the molecules related to these three things may help to organize our thoughts. 

In minor, the authors have to spell out the abbreviations in the first appearance. For example Line 118, PAS, Line 134, TBARS.

Author Response

  1. The manuscript describes PYC effect on TiO2NPs induced lung inflammation. It is overall well written and interesting. I have two major comments. It seems that Dex is more effective than PYC. I assume Dex was used as a positive control, however, there is no explanation about Dex in the main text (why Dex is examined). The authors need to explain. The other comment, the authors had better add the figure that shows the relationships of the molecules in the text, The authors examined not only inflammation but also oxidative stress and apoptosis. The results are interesting but complicated. Therefore, the figure that shows the molecules related to these three things may help to organize our thoughts.

Response: We appreciate your comments. dexamethasone is widely used as a positive control substance in respiratory efficacy studies [1-3]. Therefore, we also used dexamethasone as a positive control substance, and described in the Methods section that the dexamethasone administration group was a positive control group. In addition, as suggested, we created a Figure 9 to help understand the research results and signaling and added it to the manuscript.

References

  • Jain S, Durugkar S, Saha P, Gokhale SB, Naidu VGM, and Sharma P: Effects of intranasal azithromycin on features of cigarette smoke-induced lung inflammation. Eur J Pharmacol 915: 174467, 2022.
  • Pinheiro AJMCR, Gonçalves JS, Dourado ÁWA, de Sousa EM, Brito NM, Silva LK, Batista MCA, de Sá JC, Monteiro CRAV, Fernandes ES, Monteiro-Neto V, Campbell LA, Zago PMW, and Lima-Neto LG: Punica granatum L. leaf extract attenuates lung inflammation in mice with acute lung injury. J Immunol Res 2018: 6879183, 2018.
  • Shen H, Wu N, Wang Y, Zhao H, Zhang L, Li T, and Zhao M: Chloroquine attenuates paraquat-induced lung injury in mice by altering inflammation, oxidative stress and fibrosis. Int Immunopharmacol 46: 16–22, 2017.
  1. In minor, the authors have to spell out the abbreviations in the first appearance. For example Line 118, PAS, Line 134, TBARS.

Response: As suggested, we revised the manuscript.

Reviewer 2 Report

This manuscript presents important findings on the toxic effects of titanium dioxide nanoparticles (TiO2NPs) and the potential protective role of Pycnogenol (PYC) in pulmonary inflammation. The experiments are well-designed and the data are promising.

However, several key areas need clarification and further detail to strengthen the manuscript.

-The current title is informative but could be more concise. Consider revising it to "Pycnogenol Mitigates Titanium Dioxide Nanoparticle-Induced Lung Inflammation via Thioredoxin-Interacting Protein Downregulation.

-The Introduction lacks a comprehensive review of previous studies on TiO2NPs and PYC. Expand the literature review to include more background information and recent studies that have explored the toxicological effects of TiO2NPs and the therapeutic potential of PYC.

 -Clarify the novelty of this study in the context of existing research. What specific gaps does this study address?

-The preparation and administration of PYC, particularly its solubility in PBS and stability over the course of the experiment, require additional information.

-Clarify the rationale for choosing the specific dosages of PYC (10 mg/kg and 20 mg/kg) and dexamethasone (DEX) (1 mg/kg). Were these doses based on preliminary studies or literature references?

-The procedure for intranasal administration could benefit from more details on the technique and any measures taken to ensure uniform distribution of TiO2NPs.

-Clarify the statistical methods used for data analysis. Which tests were used? ANOVA?

-Address potential limitations of the study.

-Provide more detail on how PYC interacts with TXNIP and other signaling pathways, and discuss any potential off-target effects or toxicity of PYC itself

-Suggest specific future directions for research based on the findings of this study. For example, recommend further studies to explore the long-term effects of PYC, its efficacy in different models of lung injury, or its potential interactions with other therapeutic agents

Author Response

This manuscript presents important findings on the toxic effects of titanium dioxide nanoparticles (TiO2NPs) and the potential protective role of Pycnogenol (PYC) in pulmonary inflammation. The experiments are well-designed and the data are promising.

However, several key areas need clarification and further detail to strengthen the manuscript.

  1. The current title is informative but could be more concise. Consider revising it to "Pycnogenol Mitigates Titanium Dioxide Nanoparticle-Induced Lung Inflammation via Thioredoxin-Interacting Protein Downregulation.

Response: We appreciate your comments. We agree that the title you suggested is concise, however, we decided to keep the existing title.

  1. The Introduction lacks a comprehensive review of previous studies on TiO2NPs and PYC. Expand the literature review to include more background information and recent studies that have explored the toxicological effects of TiO2NPs and the therapeutic potential of PYC.

Response: As suggested, we supplemented the Introduction section by reviewing previous studies on TiO2NPs and PYC.

  1. Clarify the novelty of this study in the context of existing research. What specific gaps does this study address?

Response: Because of the lack of studies on the efficacy of PYC in TiO2NP-induced pulmonary toxicity and TXNIP, the study was conducted and the novelty of this study was more clearly described as suggested.

  1. The preparation and administration of PYC, particularly its solubility in PBS and stability over the course of the experiment, require additional information.

Response: Before conducting this study, we were able to find literature that provided information on the solubility and stability of PYC. According to the study by Verlaet et al. [1], the composition of PYC samples dissolved in PBS was analyzed by HPLC to confirm whether it was similar to the composition of the sample completely dissolved in methanol, and it was confirmed that there was no difference in the composition at concentrations of 1, 4, and 50 mg/mL of PYC. In addition, there was no change in the composition even after sonication for 15 minutes. Although experiments on the stability and solubility of PYC in PBS were not performed in this study, based on the above study, we set the concentration of PYC to 1 and 2 mg/mL (dosage: 10 and 20 mg/kg; volume: 10 mL/kg), and the preparation of PYC was performed immediately before daily administration, and oral administration was completed within 15 minutes.

References

  • Verlaet A, van der Bolt N, Meijer B, Breynaert A, Naessens T, Konstanti P, Smidt H, Hermans N, Savelkoul HFJ, and Teodorowicz M: Toll-like receptor-dependent immunomodulatory activity of Pycnogenol®. Nutrients 11: 214, 2019.
  1. Clarify the rationale for choosing the specific dosages of PYC (10 mg/kg and 20 mg/kg) and dexamethasone (DEX) (1 mg/kg). Were these doses based on preliminary studies or literature references?

Response: The dosages of dexamethasone [1] and pycnogenol [2] used in this study were determined based on literature references.

  • Jain S, Durugkar S, Saha P, Gokhale SB, Naidu VGM, and Sharma P: Effects of intranasal azithromycin on features of cigarette smoke-induced lung inflammation. Eur J Pharmacol 915: 174467, 2022.
  • Ko JW, Shin NR, Park SH, Kim JS, Cho YK, Kim JC, Shin IS, and Shin DH: Pine bark extract (Pycnogenol®) suppresses cigarette smoke-induced fibrotic response via transforming growth factor-β1/Smad family member 2/3 signaling. Lab Anim Res 33: 76–83, 2017.
  1. The procedure for intranasal administration could benefit from more details on the technique and any measures taken to ensure uniform distribution of TiO2NPs.

Response: As suggested, we supplemented the intranasal administration in the Methods section.

  1. Clarify the statistical methods used for data analysis. Which tests were used? ANOVA?

Response: The statistical method used in this study was ANOVA, and the manuscript was revised to clarify the statistical method.

  1. Address potential limitations of the study.

Response: As suggested, we revised the manuscript.

  1. Provide more detail on how PYC interacts with TXNIP and other signaling pathways, and discuss any potential off-target effects or toxicity of PYC itself

Response: As suggested, we revised the manuscript.

  1. Suggest specific future directions for research based on the findings of this study. For example, recommend further studies to explore the long-term effects of PYC, its efficacy in different models of lung injury, or its potential interactions with other therapeutic agents

Response: As suggested, we revised the manuscript.

Round 2

Reviewer 1 Report

I do not have further comments.

I do not have further comments.

Reviewer 2 Report

the authors covered my points

-